# Orthogonal Recurrent Neural Networks with Scaled Cayley Transform

## Abstract

Recurrent Neural Networks (RNNs) are designed to handle sequential data but suffer from vanishing or exploding gradients. Recent work on Unitary Recurrent Neural Networks (uRNNs) have been used to address this issue and in some cases, exceed the capabilities of Long Short-Term Memory networks (LSTMs). We propose a simpler and novel update scheme to maintain orthogonal recurrent weight matrices without using complex valued matrices. This is done by parametrizing with a skew-symmetric matrix using the Cayley transform. Such a parametrization is unable to represent matrices with negative one eigenvalues, but this limitation is overcome by scaling the recurrent weight matrix by a diagonal matrix consisting of ones and negative ones. The proposed training scheme involves a straightforward gradient calculation and update step. In several experiments, the proposed scaled Cayley orthogonal recurrent neural network (scoRNN) achieves superior results with fewer trainable parameters than other unitary RNNs.

## 1 Introduction

Deep neural networks have been used to solve numerical problems of varying complexity. RNNs have parameters that are reused at each time step of a sequential data point and have achieved state of the art performance on many sequential learning tasks. Nearly all optimization algorithms for neural networks involve some variant of gradient descent. One major obstacle to training RNNs with gradient descent is due to *vanishing* or *exploding* gradients, as described in Bengio et al. (1993) and Pascanu et al. (2013). This problem refers to the tendency of gradients to grow or decay exponentially in size, resulting in gradient descent steps that are too small to be effective or so large that the network oversteps the local minimum. This issue significantly diminishes RNNs' ability to learn time-based dependencies, particularly in problems with long input sequences.

A variety of architectures have been introduced to overcome this difficulty. The current preferred RNN architectures are those that introduce gating mechanisms to control when information is retained or discarded, such as LSTMs (Hochreiter & Schmidhuber, 1997) and GRUs (Cho et al., 2014), at the cost of additional trainable parameters. More recently, the unitary evolution RNN (uRNN) (Arjovsky et al., 2016) uses a parametrization that forces the recurrent weight matrix to remain unitary throughout training, and exhibits superior performance to LSTMs on a variety of synthetic and real-world tasks. For clarity, we follow the convention of Wisdom et al. (2016) and refer to this network as the restricted-capacity uRNN.

Since the introduction of uRNNs, orthogonal and unitary RNN schemes have increased in both popularity and complexity. Wisdom et al. (2016) use a multiplicative update method detailed in Tagare (2011) and Wen & Yin (2013) to expand uRNNs' capacity to include all unitary matrices. These networks are referred to as full-capacity uRNNs. Jing et al. (2016)'s EURNN parametrizes this same space with Givens rotations, while Jing et al. (2017)'s GORU introduces a gating mechanism for unitary RNNs to enable short term memory. Vorontsov et al. (2017) introduced modified optimization and regularization methods that restrict singular values of the recurrent matrix to an interval around 1. Each of these methods involve complex valued recurrent weights. For other work in addressing the vanishing and exploding gradient problem, see Henaff et al. (2017) and Le et al. (2015).

In this paper, we consider RNNs with a recurrent weight matrix taken from the set of all orthogonal matrices. To construct the orthognal weight matrix, we parametrize it with a skew-symmetric matrix through a scaled Cayley transform. This scaling allows us to avoid the singularity issue occuring

for $-1$ eigenvalues that may arise in the standard Cayley transform. With the parameterization, the network optimization involves a relatively simple gradient descent update. The resulting method achieves superior performance on sequential data tasks with a smaller number of trainable parameters and hidden sizes than other unitary RNNs and LSTMs.

The method we present in this paper works entirely with real matrices, and as such, our results deal only with orthogonal and skew-symmetric matrices. However, the method and all related theory remain valid for unitary and skew-Hermitian matrices in the complex case. The experimental results in this paper indicate that state of the art performance can be achieved without the increased complexity of optimization along the Stiefel manifold and using complex matrices.

## 2 BACKGROUND

### 2.1 RECURRENT NEURAL NETWORKS

A recurrent neural network (RNN) is a function with input parameters $U \in \mathbb{R}^{n \times m}$, recurrent parameters $W \in \mathbb{R}^{n \times n}$, recurrent bias $b \in \mathbb{R}^n$, output parameters $V \in \mathbb{R}^{p \times n}$, and output bias $c \in \mathbb{R}^p$ where $m$ is the data input size, $n$ is the number of hidden units, and $p$ is the output data size. From an input sequence $x = (x_1, x_2, ..., x_T)$ where $x_i \in \mathbb{R}^m$, the RNN returns an output sequence $y = (y_1, y_2, ..., y_T)$ where each $y_i \in \mathbb{R}^p$ is given recursively by

$$h_t = \sigma \left( U x_t + W h_{t-1} + b \right)$$
$$y_t = V h_t + c$$

where $h = (h_0, \ldots, h_{T-1})$, $h_i \in \mathbb{R}^n$ is the hidden layer state at time $i$ and $\sigma(\cdot)$ is the activation function, which is often a pointwise nonlinearity such as a hyperbolic tangent function or rectified linear unit (Nair & Hinton, 2010).

### 2.2 UNITARY RNNs

A real matrix $W$ is orthogonal if it satisfies $W^T W = I$. The complex analog of orthogonal matrices are unitary matrices, which satisfy $W^* W = I$, where $*$ denotes the conjugate transpose. Orthogonal and unitary matrices have the desirable property that $\|Wx\|_2 = \|x\|_2$ for any vector $x$. This property motivates the use of orthogonal or unitary matrices in RNNs to avoid vanishing and exploding gradients, as detailed in Arjovsky et al. (2016).

Arjovsky et al. (2016) follow the framework of the previous section for their restricted-capacity uRNN, but introduce a parametrization of the recurrent matrix $W$ using a product of simpler matrices. This parameterization is given by a product consisting of diagonal matrices with complex norm 1, complex Householder reflection matrices, discrete Fourier transform matrices, and a fixed permutation matrix with the resulting product being unitary.

Wisdom et al. (2016) note that this representation has only $7n$ parameters, which is insufficient to represent all unitary matrices for $n > 7$. In response, they present the full-capacity uRNN, which uses a multiplicative update step that is able to reach all unitary matrices of order $n$.

The full-capacity uRNN aims to construct a unitary matrix $W^{(k+1)}$ from $W^{(k)}$ by moving along a curve on the Stiefel manifold $\{W \in \mathbb{C}^{n \times n} \mid W^* W = I\}$. For the network optimization, it is necessary to use a curve that is in a descent direction of the cost function $L := L(W)$. In Tagare (2011), Wen & Yin (2013), and Wisdom et al. (2016), a descent direction is constructed as $B^{(k)} W^{(k)}$, which is a representation of the derivative operator $DL(W^{(k)})$ in the tangent space of the Stiefel manifold at $W^{(k)}$. Then, with $B^{(k)} W^{(k)}$ defining the direction of a descent curve, an update along the Stiefel manifold is obtained as

$$W^{(k+1)} = \left( I + \frac{\lambda}{2} B^{(k)} \right)^{-1} \left( I - \frac{\lambda}{2} B^{(k)} \right) W^{(k)} \tag{1}$$

where $\lambda$ is the learning rate.

## 3 SCALED CAYLEY ORTHOGONAL RNN

### 3.1 CAYLEY TRANSFORM

The Cayley transform gives a bijection between the set of orthogonal matrices without $-1$ eigenvalues and the set of skew-symmetric matrices (i.e., matrices where $A^T = -A$):

$$W = (I + A)^{-1}(I - A), \qquad A = (I + W)^{-1}(I - W).$$

We can use this bijection to parametrize the set of orthogonal matrices without $-1$ eigenvalues using skew-symmetric matrices. This parametrization is attractive from a machine learning perspective because it is closed under addition: the sum or difference of two skew-symmetric matrices is also skew-symmetric, so we can use gradient descent algorithms like RMSprop (Tieleman & Hinton, 2012) or Adam (Kingma & Ba, 2014) to train parameters.

However, this parametrization cannot represent orthogonal matrices with $-1$ eigenvalues, since in this case $I + W$, is not invertible. Theoretically, we can still represent matrices with eigenvalues that are arbitrarily close to $-1$; however, it can require large entries of $A$. For example, a 2x2 orthogonal matrix $W$ with eigenvalues $\approx -0.99999 \pm 0.00447i$ and its parametrization $A$ by the Cayley transform is given below.

$$W = \begin{bmatrix} -0.99999 & -\sqrt{1 - 0.99999^2} \\ \sqrt{1 - 0.99999^2} & -0.99999 \end{bmatrix} \qquad A \approx \begin{bmatrix} 0 & 447.212 \\ -447.212 & 0 \end{bmatrix}$$

Gradient descent algorithms will learn this $A$ matrix very slowly, if at all. This difficulty can be overcome through a suitable diagonal scaling according to results from Kahan (2006).

**Theorem 3.1** *Every orthogonal matrix $W$ can be expressed as*

$$W = (I + A)^{-1}(I - A)D$$

*where $A = [a_{ij}]$ is real-valued, skew-symmetric with $|a_{ij}| \leq 1$, and $D$ is diagonal with all nonzero entries equal to $\pm 1$.*

We call the transform in Theorem 3.1 the scaled Cayley transform. Then, with an appropriate choice of $D$, the scaled Cayley transform can reach any orthogonal matrix including those with $-1$ eigenvalues. Further, it ensures that the skew-symmetric matrix $A$ that generates the orthogonal matrix will be bounded.

Our proposed network, the scaled Cayley orthogonal recurrent neural network (scoRNN), is based on this theorem. We parametrize the recurrent weight matrix $W$ through a skew-symmetric matrix $A$, which results in $\frac{n(n-1)}{2}$ trainable weights. The recurrent matrix $W$ is formed by the scaled Cayley transform: $W = (I + A)^{-1}(I - A)D$. The scoRNN then operates identically to the set of equations given in Section 2.1, but during training we update the skew-symmetric matrix $A$ using gradient descent, while $D$ is fixed throughout the training process. The number of $-1$s on the diagonal of $D$, which we call $\rho$, is considered a hyperparameter in this work and is manually chosen based on the task.

### 3.2 UPDATE SCHEME

To update the recurrent parameter matrix $A$ as described in Section 3.1, we must find the gradients of $A$ by backpropagating through the Cayley transform. The following theorem describes these gradients. A proof is given in Appendix A.

**Theorem 3.2** *Let $L = L(W) : \mathbb{R}^{n \times n} \to \mathbb{R}$ be some differentiable loss function for an RNN with the recurrent weight matrix $W$. Let $W = W(A) := (I + A)^{-1}(I - A)D$ where $A \in \mathbb{R}^{n \times n}$ is skew-symmetric and $D \in \mathbb{R}^{n \times n}$ is a fixed diagonal matrix consisting of -1 and 1 entries. Then the gradient of $L = L(W(A))$ with respect to $A$ is*

$$\frac{\partial L}{\partial A} = V^T - V \tag{2}$$

*where $V := (I + A)^{-T} \frac{\partial L}{\partial W} (D + W^T)$, $\frac{\partial L}{\partial A} = \left[ \frac{\partial L}{\partial A_{i,j}} \right] \in \mathbb{R}^{n \times n}$, and $\frac{\partial L}{\partial W} = \left[ \frac{\partial L}{\partial W_{i,j}} \right] \in \mathbb{R}^{n \times n}$.*

At each training step of scoRNN, we first use the standard backpropagation algorithm to compute $\frac{\partial L}{\partial W}$ and then use Theorem 3.2 to compute $\frac{\partial L}{\partial A}$. We then update $A$ with gradient descent (or a related optimization method), and reconstruct $W$ as follows:

$$A^{(k+1)} = A^{(k)} - \lambda \frac{\partial L(W(A^{(k)}))}{\partial A}$$
$$W^{(k+1)} = \left(I + A^{(k+1)}\right)^{-1} \left(I - A^{(k+1)}\right) D$$

The skew-symmetry of $\frac{\partial L}{\partial A}$ ensures that $A^{(k+1)}$ will be skew-symmetric and, in turn, $W^{(k+1)}$ will be orthogonal.

The scoRNN and the full-capacity uRNN from Section 2.2 both have the capacity to optimize an orthogonal or unitary recurrent matrix $W$, but they use different update schemes. The full-capacity uRNN performs a multiplicative update that moves $W$ along the tangent space of the Stiefel manifold, which can be shown to be a descent direction, but not necessarily the steepest one. In contrast, scoRNN performs an additive update in the direction of steepest descent with respect to its parametrization. The scoRNN update proves to be much more resistant to loss of orthogonality during training; see Appendix B. It also maintains stable hidden state gradients in the sense that the gradient norm does not change significantly in time; see Appendix C for experimental results. This is achieved with very little overhead computational costs over the standard RNN; see Appendix D for experiments comparing computational speeds.

## 4 OTHER ARCHITECTURE DETAILS

The basic architecture of scoRNN is very similar to the standard RNN as presented in Section 2.1. From a network layer perspective, one can think of the application of the recurrent weight in a three layer process. Let $h_t \in \mathbb{R}^n$ be the current state of the scoRNN at a particular time step, $t$. We then pass $h_t$ through the following layers:

- Layer 1: $h_t \to D h_t =: h_t^{(1)}$
- Layer 2: $h_t^{(1)} \to (I - A) h_t^{(1)} =: h_t^{(2)}$
- Layer 3: $h_t^{(2)} \to (I + A)^{-1} h_t^{(2)} =: h_t^{(3)}$

Note that the above scheme is the same as taking $h_t \to W h_t$ as discussed previously.

### 4.1 MODRELU ACTIVATION FUNCTION

The modReLU function was first implemented by Arjovsky et al. (2016) to handle complex valued functions and weights. Unlike previous methods, our method only uses real-valued functions and weights. Nevertheless, we have found that the modReLU function in the real case also performed better than other activation functions. The function is defined as

$$\sigma_{\text{modReLU}}(z) = \frac{z}{|z|} \sigma_{\text{ReLU}}\left(|z| + b\right) = \begin{cases} \frac{z}{|z|}\left(|z| + b\right) & \text{if } |z| + b \geq 0 \\ 0 & \text{if } |z| + b < 0 \end{cases} \tag{3}$$

where $b$ is a trainable bias. In the real case, this simplifies to $\text{sign}(z)\sigma_{\text{ReLU}}(|z| + b)$. To implement this activation function in scoRNN, we replace the computation of $h_t$ in Section 2.1 with

$$z_t = U x_t + W h_{t-1}$$
$$h_t = \sigma_{\text{modReLU}}(z_t)$$

We believe that the improved performance of the modReLU over other activation functions, such as ReLU, is because it admits both positive and negative activation values, which appears to be important for the state transition in orthogonal RNNs. This is similar to the hyperbolic tangent function but does not have vanishing gradient issues.

### 4.2 INITIALIZATION

Modifying the initialization of our parameter matrices, in particular our recurrent parameter matrix $A$, had a significant effect on performance. The most effective initialization method we found uses

a technique inspired by Henaff et al. (2017). We initialize all of the entries of $A$ to be 0 except for 2x2 blocks along the diagonal, which are given as

$$A = \begin{bmatrix} B_1 & & \\ & \ddots & \\ & & B_{\lfloor n/2 \rfloor} \end{bmatrix} \quad \text{where} \quad B_j = \begin{bmatrix} 0 & s_j \\ -s_j & 0 \end{bmatrix}$$

with $s_j = \sqrt{\frac{1-\cos(t_j)}{1+\cos(t_j)}}$ and $t_j$ is sampled uniformly from $\left[0, \frac{\pi}{2}\right]$. The Cayley transform of this $A$ will have eigenvalues equal to $\pm e^{it_j}$ for each $j$, which will be distributed uniformly along the right unit half-circle. Multiplication by the scaling matrix $D$ will reflect $\rho$ of these eigenvalues across the imaginary axis. We use this method to initialize scoRNN's $A$ matrix in all of the experiments listed in section 5.

## 5 EXPERIMENTS

For each experiment, we found optimal hyperparameters for scoRNN using a grid search. For other models, we used the best hyperparameters settings as reported in Wisdom et al. (2016) and Arjovsky et al. (2016). If not available, we performed a grid search to find the best hyperparameters.

### 5.1 COPYING PROBLEM

This experiment follows descriptions found in Arjovsky et al. (2016) and Wisdom et al. (2016), and tests an RNN's ability to reproduce a sequence seen many timesteps earlier. In the problem setup, there are 10 input classes, which we denote using the digits 0-9, with 0 being used as a 'blank' class and 9 being used as a 'marker' class. The RNN receives an input sequence of length $T + 20$. This sequence consists of entirely zeros, except for the first ten elements, which are uniformly sampled from classes 1-8, and a 9 placed ten timesteps from the end. The goal for the machine is to output zeros until it sees a 9, at which point it should output the ten elements from the beginning of the input sequence. Thus, information must propagate from the beginning to the end of the sequence for a machine to successfully learn this task, making it critical to avoid vanishing/exploding gradients.

A baseline strategy with which to compare machine performance is that of outputting 0 until the machine sees a 9, and then outputting 10 elements randomly sampled from classes 1-8. The expected cross-entropy for such a strategy is $\frac{10 \log(8)}{T+20}$. In practice, it is common to see gated RNNs such as LSTMs converge to this local minimum.

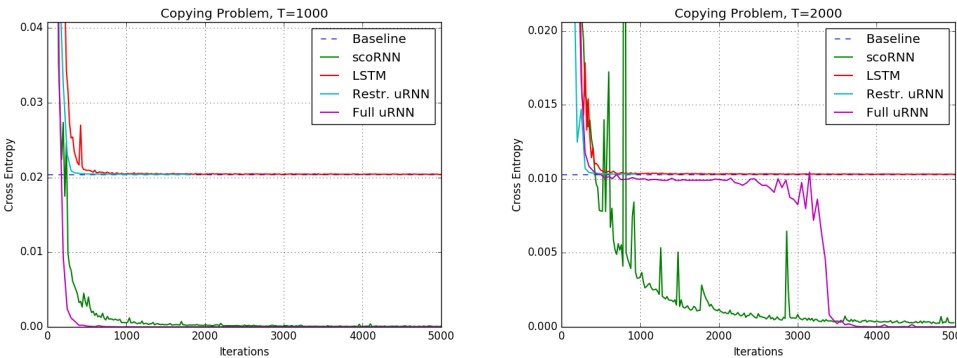

Figure 1: Cross entropy of each machine on the copying problem with $T = 1000$ (left) and $T = 2000$ (right).

We vary the number of hidden units of the machines to match the number of parameters, approximately 22k each. This results in an LSTM with $n = 68$, a restricted-capacity uRNN with $n = 470$, a full-capacity uRNN with $n = 128$, and a scoRNN with $n = 190$. We found the best performance with the scoRNN came from $\rho = n/2$, which gives an initial $W$ with eigenvalues distributed uniformly on the entire unit circle.

Figure 1 compares each model's performance for $T = 1000$ and $T = 2000$, with the baseline cross-entropy given as a dashed line. In both cases, cross-entropy for the restricted-capacity uRNN and LSTM never drop below the baseline. For the $T = 1000$ test, the full-capacity uRNN and scoRNN converge immediately to zero entropy solutions, with the full-capacity uRNN converging slightly faster. For $T = 2000$, the full-capacity uRNN remains at the baseline for several thousand iterations, but is eventually able to find a correct solution. In contrast, the scoRNN error has a smooth convergence that bypasses the baseline, but does so more slowly than the full-capacity uRNN.

## 5.2 ADDING PROBLEM

We examined a variation of the adding problem as proposed by Arjovsky et al. (2016) which is based on the work of Hochreiter & Schmidhuber (1997). This variation involves passing two sequences concurrently into the RNN, each of length $T$. The first sequence is a sequence of digits sampled uniformly with values ranging in a half-open interval, $\mathcal{U}[0, 1)$. The second sequence is a marker sequence consisting of all zeros except for two entries that are marked by one. The first 1 is located uniformly within the interval $[1, \frac{T}{2})$ of the sequence and the second 1 is located uniformly within the interval $[\frac{T}{2}, T)$ of the sequence. The label for each pair of sequences is the sum of the two entries that are marked by one, which forces the machine to identify relevant information in the first sequence among noise. As the sequence length increases, it becomes more crucial to avoid vanishing/exploding gradients. Naively predicting one regardless of the sequence gives an expected mean squared error (MSE) of approximately 0.167. This will be considered as the baseline.

The number of hidden units for each network was adjusted so that each had approximately 14k trainable parameters. This results in $n = 170$ for the scoRNN, $n = 60$ for the LSTM, $n = 120$ for the Full-Capacity uRNN, and $n = 950$ hidden units for the restricted-capacity uRNN. The test set

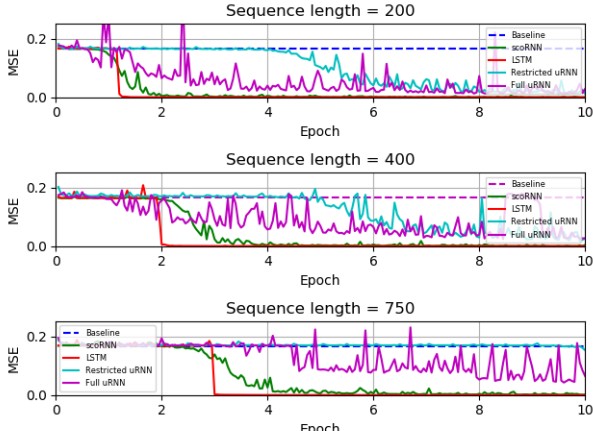

Figure 2: Test set MSE for each machine on the adding problem with sequence lengths of $T = 200$ (top), $T = 400$ (middle), and $T = 750$ (bottom).

MSE results for sequence lengths $T = 200$, $T = 400$, and $T = 750$ can be found in Figure 2. A training set size of 100,000 and a testing set size of 10,000 were used for each sequence length. For each case, the networks start at or near the baseline MSE and drop towards zero after a few epochs. As the sequence length increases, the number of epochs before the drop increases. We found the best settings for the scoRNN were $\rho = n/2$ for $T = 200$ and $\rho = 7n/10$ for $T = 400$ and $T = 750$. As can be seen, the LSTM error drops precipitously across the board before all other models, while the unitary and orthogonal RNNs descend more gradually. Although in some cases the full-capacity uRNN begins to drop below the baseline before scoRNN, the full-capacity uRNN does not drop as quickly and has a more irregular descent curve.

## 5.3 Pixel-by-Pixel MNIST

We ran two experiments based around classifying samples from the well-known MNIST dataset (Le-Cun et al.). Following the implementation of Le et al. (2015), each pixel of the image is fed into the RNN sequentially, resulting in a single pixel sequence length of 784. In the first experiment, which we refer to as unpermuted MNIST, pixels are arranged in the sequence row-by-row. In the second, which we call permuted MNIST, a fixed permutation is applied to training and testing sequences.

All scoRNN machines were trained with the RMSProp optimization algorithm. Input and output weights used a learning rate of $10^{-3}$, while the recurrent parameters used a learning rate of $10^{-4}$ (for $n = 170$) or $10^{-5}$ (for $n = 360$ and $n = 512$). For unpermuted MNIST, we found $\rho$ to be optimal at $n/10$, while the best value of $\rho$ for permuted MNIST was $n/2$. We suspect that the difference of these two values comes from the different types of dependencies in each: unpermuted MNIST has mostly local dependencies, while permuted MNIST requires learning many long-term dependencies, which appear to be more easily modeled when the diagonal of $D$ has a higher proportion of $-1$s.

Each experiment used a training set of 55,000 images and a test set of 10,000 testing images. Each machine was trained for 70 epochs, and test set accuracy, the percentage of test images classified correctly, was evaluated at the conclusion of each epoch. Figure 3 shows test set accuracy over time for each machine, and the best performance over all epochs by each machine is given in Table 1.

Table 1: Results for unpermuted and permuted pixel-by-pixel MNIST experiments. Evaluation accuracies are based on the best test accuracy at the end of every epoch.

| Model | n | # parameters | MNIST Test Accuracy | Permuted MNIST Test Accuracy |
|---|---|---|---|---|
| scoRNN | 170 | $\approx 16$k | 0.973 | 0.943 |
| scoRNN | 360 | $\approx 69$k | 0.983 | 0.962 |
| scoRNN | 512 | $\approx 137$k | 0.985 | 0.966 |
| LSTM | 128 | $\approx 68$k | 0.987 | 0.920 |
| LSTM | 256 | $\approx 270$k | 0.989 | 0.929 |
| LSTM | 512 | $\approx 1,058$k | 0.985 | 0.920 |
| Restricted-capacity uRNN | 512 | $\approx 16$k | 0.976 | 0.945 |
| Restricted-capacity uRNN | 2170 | $\approx 69$k | 0.984 | 0.953 |
| Full-capacity uRNN | 116 | $\approx 16$k | 0.947 | 0.925 |
| Full-capacity uRNN | 512 | $\approx 270$k | 0.974 | 0.947 |

In both experiments, the 170 hidden unit scoRNN gives similar performance to both of the 512 hidden unit uRNNs using a much smaller hidden dimension and, in the case of the full-capacity uRNN, an order of magnitude fewer parameters. Matching the number of parameters ($\approx 69k$), the 2170 restricted-capacity uRNN performance was comparable to the 360 hidden unit scoRNN for unpermuted MNIST, but performed worse for permuted MNIST, and required a much larger hidden size and a significantly longer run time, see Appendix D. As in experiments presented in Arjovsky et al. (2016) and Wisdom et al. (2016), orthogonal and unitary RNNs are unable to outperform the LSTM in the unpermuted case. However, the 360 and 512 hidden unit scoRNNs outperform the unitary RNNs. On permuted MNIST, the 512 hidden unit scoRNN achieves a test-set accuracy of 96.6%, outperforming all of the uRNNs and LSTMs. We believe this is a state of the art result.

## 5.4 TIMIT Speech Dataset

To see how the models performed on audio data, speech prediction was performed on the TIMIT dataset (Garofolo et al., 1993), a collection of real-world speech recordings. Excluding the dialect SA sentences and using only the core test set, the dataset consisted of 3,696 training and 192 testing audio files. Similar to experiments in Wisdom et al. (2016), audio files were downsampled to 8kHz and a short-time Fourier transform (STFT) was applied with a Hann window of 256 samples and a window hop of 128 samples (16 milliseconds). The result is a set of frames, each with 129 complex-valued Fourier amplitudes. The log-magnitude of these amplitudes is used as the input data for the machines. Each frame was fed into the machine sequentially, and at each time step $t$, the machine's

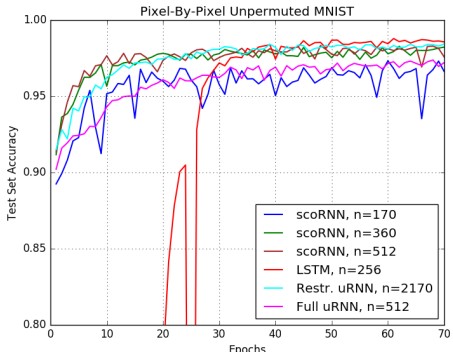 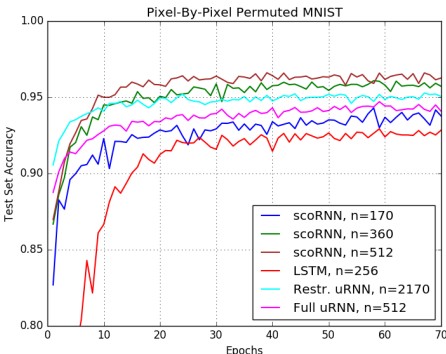

Figure 3: Test accuracy for unpermuted and permuted MNIST over time. All scoRNN models and only the best performing models for each other architectures are shown.

target output is to predict the $t+1$ frame. For each model, the hidden layer sizes were adjusted such that each model had approximately equal numbers of trainable parameters. For scoRNN, we used the Adam optimizer with learning rate $10^{-3}$ to train the input and output parameters, and RMSprop with a learning rate of $10^{-3}$ (for $n = 224$) or $10^{-4}$ (for $n = 322$ and $n = 425$) to train the recurrent weight matrix. The number of negative eigenvalues used was $\rho = n/10$.

The loss function used for training was the mean squared error (MSE) between the predicted and actual log-magnitudes of the next time frame over the entire sequence. Table 2 contains the MSE on validation and testing sets, which shows that all scoRNN models achieve a smaller MSE than all LSTM and unitary RNN models. Similar to Wisdom et al. (2016), we reconstructed audio files using the predicted log-magnitudes from each machine and evaluated them on several audio metrics. We found that the scoRNN predictions achieved better scores on the signal-to-noise ratio metric SegSNR (Brookes et al., 1997), but performed slightly worse than the full-capacity uRNN predictions on the human intelligibility and perception metrics STOI (Taal et al., 2011) and PESQ (Rix et al., 2001).

Table 2: Results for the TIMIT speech dataset. Evaluation based on MSE and various audio metrics

| Model | n | # params | Valid. MSE | Eval. MSE | Model | n | # params | Valid. MSE | Eval. MSE |
|-------|---|----------|------------|-----------|-------|---|----------|------------|-----------|
| scoRNN | 224 | $\approx 83$k | 9.26 | 8.50 | Rest. uRNN | 158 | $\approx 83$k | 15.57 | 18.51 |
| scoRNN | 322 | $\approx 135$k | 8.48 | 7.82 | Rest. uRNN | 256 | $\approx 135$k | 15.90 | 15.31 |
| scoRNN | 425 | $\approx 200$k | 7.97 | 7.36 | Rest. uRNN | 378 | $\approx 200$k | 16.00 | 15.15 |
| LSTM | 84 | $\approx 83$k | 18.43 | 17.18 | Full uRNN | 128 | $\approx 83$k | 15.07 | 14.58 |
| LSTM | 120 | $\approx 135$k | 17.05 | 15.91 | Full uRNN | 192 | $\approx 135$k | 15.10 | 14.50 |
| LSTM | 158 | $\approx 200$k | 16.33 | 16.06 | Full uRNN | 256 | $\approx 200$k | 14.96 | 14.69 |

# 6 CONCLUSION

There have been recent breakthroughs with RNN architectures using unitary recurrent weight matrices to address the vanishing/exploding gradient problem. These unitary RNNs are implemented with complex valued matrices and require additional complexity in computation. Unlike unitary RNNs, the scoRNN developed in this paper uses real valued orthogonal recurrent weight matrices with a simpler implementation scheme by parametrizing with a skew-symmetric matrix. The resulting model's additive update step is in the direction of steepest descent with respect to this parametrization, and maintains the orthogonality of the recurrent weight matrix in the presence of roundoff errors. Results from our experiments show that scoRNN can achieve superior performance to unitary RNNs, in some cases with many fewer trainable parameters than other models.

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

## APPENDIX A: PROOF OF THEOREM 3.2

For completeness, we restate and prove Theorem 3.2.

**Theorem 3.2** *Let $L = L(W) : \mathbb{R}^{n \times n} \to \mathbb{R}$ be some differentiable loss function for an RNN with the recurrent weight matrix $W$. Let $W = W(A) := (I + A)^{-1} (I - A) D$ where $A \in \mathbb{R}^{n \times n}$ is skew-symmetric and $D \in \mathbb{R}^{n \times n}$ is a fixed diagonal matrix consisting of -1 and 1 entries. Then the gradient of $L = L(W(A))$ with respect to $A$ is*

$$\frac{\partial L}{\partial A} = V^T - V \tag{4}$$

*where $V := (I + A)^{-T} \frac{\partial L}{\partial W} (D + W^T)$, $\frac{\partial L}{\partial A} = \left[ \frac{\partial L}{\partial A_{i,j}} \right] \in \mathbb{R}^{n \times n}$, and $\frac{\partial L}{\partial W} = \left[ \frac{\partial L}{\partial W_{i,j}} \right] \in \mathbb{R}^{n \times n}$*

**Proof:** Let $Z := (I + A)^{-1}(I - A)$. We consider the $(i, j)$ entry of $\frac{\partial L}{\partial A}$. Taking the derivative with respect to $A_{i,j}$ where $i \neq j$ we obtain:

$$\frac{\partial L}{\partial A_{i,j}} = \sum_{k,l=1}^{n} \frac{\partial L}{\partial W_{k,l}} \frac{\partial W_{k,l}}{\partial A_{i,j}} = \sum_{k,l=1}^{n} \frac{\partial L}{\partial W_{k,l}} D_{l,l} \frac{\partial Z_{k,l}}{\partial A_{i,j}} = \text{tr} \left[ \left( \frac{\partial L}{\partial W} D \right)^T \frac{\partial Z}{\partial A_{i,j}} \right]$$

Using the identity $(I + A) Z = I - A$ and taking the derivative with respect to $A_{i,j}$ to both sides we obtain:

$$\frac{\partial Z}{\partial A_{i,j}} + \frac{\partial A}{\partial A_{i,j}} Z + A \frac{\partial Z}{\partial A_{i,j}} = -\frac{\partial A}{\partial A_{i,j}}$$

and rearranging we get:

$$\frac{\partial Z}{\partial A_{i,j}} = -(I + A)^{-1} \left( \frac{\partial A}{\partial A_{i,j}} + \frac{\partial A}{\partial A_{i,j}} Z \right)$$

Let $E_{i,j}$ denote the matrix whose $(i, j)$ entry is 1 with all others being 0. Since $A$ is skew-symmetric, we have $\frac{\partial A}{\partial A_{i,j}} = E_{i,j} - E_{j,i}$. Combining everything, we have:

$$\begin{aligned}
\frac{\partial L}{\partial A_{i,j}} &= -\text{tr} \left[ \left( \frac{\partial L}{\partial W} D \right)^T (I + A)^{-1} (E_{i,j} - E_{j,i} + E_{i,j} Z - E_{j,i} Z) \right] \\
&= -\text{tr} \left[ \left( \frac{\partial L}{\partial W} D \right)^T (I + A)^{-1} E_{i,j} \right] + \text{tr} \left[ \left( \frac{\partial L}{\partial W} D \right)^T (I + A)^{-1} E_{j,i} \right] \\
&\quad - \text{tr} \left[ \left( \frac{\partial L}{\partial W} D \right)^T (I + A)^{-1} E_{i,j} Z \right] + \text{tr} \left[ \left( \frac{\partial L}{\partial W} D \right)^T (I + A)^{-1} E_{j,i} Z \right] \\
&= -\left[ \left( \left( \frac{\partial L}{\partial W} D \right)^T (I + A)^{-1} \right)^T \right]_{i,j} + \left[ \left( \frac{\partial L}{\partial W} D \right)^T (I + A)^{-1} \right]_{i,j} \\
&\quad - \left[ \left( \left( \frac{\partial L}{\partial W} D \right)^T (I + A)^{-1} \right)^T Z^T \right]_{i,j} + \left[ Z \left( \frac{\partial L}{\partial W} D \right)^T (I + A)^{-1} \right]_{i,j}
\end{aligned}$$

$$= \left[ (I+Z)\left(\frac{\partial L}{\partial W}D\right)^T (I+A)^{-1} \right]_{i,j} - \left[ \left( \left(\frac{\partial L}{\partial W}D\right)^T (I+A)^{-1} \right)^T (I+Z^T) \right]_{i,j}$$

$$= \left[ (D+W)\left(\frac{\partial L}{\partial W}\right)^T (I+A)^{-1} \right]_{i,j} - \left[ (I+A)^{-T}\frac{\partial L}{\partial W}(D+W^T) \right]_{i,j}$$

Using the above formulation, $\frac{\partial L}{\partial A_{j,j}} = 0$ and $\frac{\partial L}{\partial A_{i,j}} = -\frac{\partial L}{\partial A_{j,i}}$ so that $\frac{\partial L}{\partial A}$ is a skew-symmetric matrix. Finally, by the definition of $V$ we get the desired result. ∎

## APPENDIX B: LOSS OF ORTHOGONALITY

In the scoRNN architecture, the recurrent weight matrix is parameterized with a skew-symmetric matrix through the Cayley transform. This ensures the computed recurrent weight matrix in floating point arithmetic is orthogonal to the order of machine precision after each update step. Unlike scoRNN, the full-capacity uRNN maintains a unitary recurrent weight matrix by a multiplicative update scheme. Due to the accumulation of rounding errors over a large number of repeated matrix multiplications, the recurrent weight may not remain unitary throughout training. To investigate this, we ran the scoRNN and full-capacity uRNN with equal hidden unit sizes of $n = 512$ on the unpermuted MNIST experiment and checked for loss of orthogonality at each epoch. The results of this experiment are shown in Figure 4. As can be seen, the recurrent weight matrix for the full-capacity uRNN becomes less unitary over time, but the orthogonality recurrent weight matrix for scoRNN is not affected by roundoff errors.

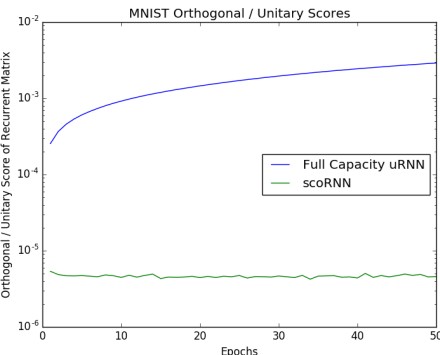

Figure 4: Unitary scores $\left(\|W^*W - I\|_F\right)$ for the full-capacity uRNN recurrent weight matrix and orthogonality scores $\left(\|W^TW - I\|_F\right)$ for the scoRNN recurrent weight matrix using a GPU on the pixel-by-pixel MNIST experiment.

## APPENDIX C: VANISHING GRADIENTS

As discussed in Arjovsky et al. (2016), the vanishing/exploding gradient problem is caused by rapid growth or decay of the gradient of the hidden state $\frac{\partial L}{\partial h_t}$ as we move earlier in the sequence (that is, as $t$ decreases). To see if vanishing/exploding gradients affect the scoRNN model, we examined hidden state gradients in the scoRNN and LSTM models on the adding problem experiment (see section 5.2) with sequence length $T = 500$.

The norms of these gradients are shown at two different points in time during training in Figure 5. As can be seen, LSTM gradient norms decrease steadily as we move away from the end of the sequence. The right half of Figure 5 shows that this vanishing effect is exacerbated by training after 300 iterations.

In contrast, scoRNN gradients decay by less than an order of magnitude at the beginning of training, remaining near $10^{-2}$ for all timesteps. Even after 300 iterations of training, scoRNN hidden state

gradients decay only slightly, from $10^{-3}$ at $t = 500$ to $10^{-4}$ at $t = 0$. This allows information to easily propagate from the beginning of the sequence to the end.

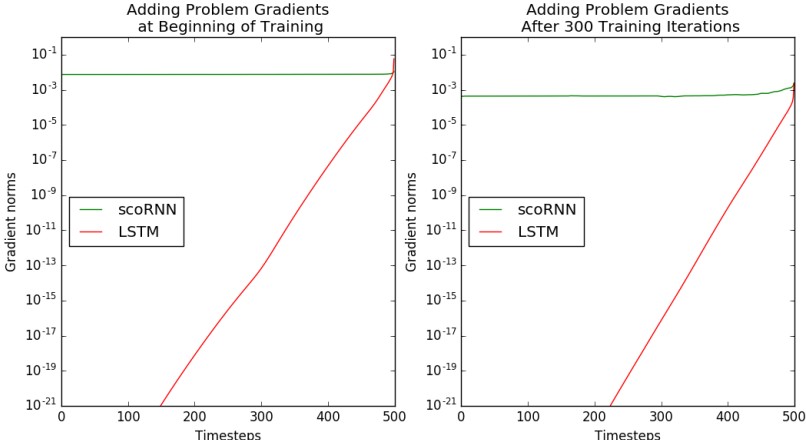

Figure 5: Gradient norms $\|\frac{\partial L}{\partial h_t}\|$ for scoRNN and LSTM models during training on the adding problem. The $x$-axis shows different values of $t$. The left plot shows gradients at the beginning of training, and the right shows gradients after 300 training iterations.

## APPENDIX D: COMPLEXITY AND SPEED

The scoRNN architecture is similar in complexity and memory usage to a standard RNN except for the additional memory requirement of storing the $n(n-1)/2$ entries of the skew-symmetric matrix $A$ and the additional complexity of forming the recurrent weight matrix $W$ from $A$ with the scaled Cayley transform. We note that the recurrent weight matrix is generated from the skew-symmetric $A$ matrix only once per training iteration; this computational cost is small compared to the cost of the forward and backward propagations through all time steps in a training batch for a standard RNN.

Table 3: Timing results for the unpermuted MNIST dataset.

| Model | n | # params | Minutes Per Epoch |
|---|---|---|---|
| scoRNN | 170 | $\approx 16$k | 5.3 |
| Rest. uRNN | 512 | $\approx 16$k | 8.2 |
| Full uRNN | 116 | $\approx 16$k | 10.8 |
| LSTM | 128 | $\approx 68$k | 5.0 |
| scoRNN | 360 | $\approx 69$k | 7.4 |
| Rest. uRNN | 2,170 | $\approx 69$k | 50.1 |
| scoRNN | 512 | $\approx 137$k | 11.2 |
| Full uRNN | 360 | $\approx 137$k | 25.8 |
| LSTM | 256 | $\approx 270$k | 5.2 |
| Full uRNN | 512 | $\approx 270$k | 27.9 |
| LSTM | 512 | $\approx 1,058$k | 5.6 |

To experimentally quantify potential differences between scoRNN and the other models, the real run-time for the models on the unpermuted MNIST experiment were recorded and are included in Table 3. All models were run on the same machine, which has an Intel Core i5-7400 processor and an nVidia GeForce GTX 1080 GPU. The scoRNN and LSTM models were run in Tensorflow, while the full and restricted capacity uRNNs were run using code provided in Wisdom et al. (2016).

The LSTM model was fastest, and hidden sizes largely did not affect time taken per epoch; we suspect this is because the LSTM model we used was built in to Tensorflow. The LSTMs are of

simialr speed to the $n = 170$ scoRNN, while they are approximately twice as fast as the $n = 512$ scoRNN. Matching the number of hidden parameters, the scoRNN model with $n = 170$ is approximately 1.5 times faster than the restricted-capacity uRNN with $n = 512$, and twice as fast as the full-capacity uRNN with $n = 116$. This relationship can also be seen in the scoRNN and full-capacity uRNN models with $\approx 137k$ parameters, where the scoRNN takes 11.2 minutes per epoch as compared to 25.8 minutes for the scoRNN.

