# OpenReview forum: "Orthogonal Recurrent Neural Networks with Scaled Cayley Transform"
_ICLR.cc/2018/Conference — Reject_

### Official Review · AnonReviewer2 · 2017-11-27
**Review of "Orthogonal Recurrent Neural Networks with Scaled Cayley Transform"**

**Rating:** 7
**Confidence:** 3

**Review:**

This manuscript introduce a scheme for learning the recurrent parameter matrix in a neural network that uses the Cayley transform and a scaling weight matrix. This scheme leads to good performance on sequential data tasks and requires fewer parameters than other techniques

Comments:
-- It’s not clear to me how D is determined for each test. Given the definition in Theorem 3.1 it seems like you would have to have some knowledge of how many eigenvalues in W you expect to be close to -1.
-- For the copying and adding problem test cases, it might be useful to clarify or cite something clarifying that the failure mode RNNs run into with temporal ordering problems is an exploding gradient, rather than any other pathological training condition, just to make it clear why these experiments are relevant.
-- The ylabel in Figure 1 is “Test Loss” which I didn’t see defined. Is this test loss the cross entropy? If so, I think it would be more effective to label the plot with that.
-- The plots in figure 1 and 2 have different colors to represent the same set of techniques. I would suggest keeping a  consistent color scheme
-- It looks like in Figure 1 the scoRNN is outperformed by the uRNN in the long run in spite of the scoRNN convergence being smoother, which should be clarified.
-- It looks like in Figure 2 the scoRNN is outperformed by the LSTM across the board, which should be clarified.
-- How is test set accuracy defined in section 5.3? Classifying digits? Recreating digits?
-- When discussing table 1, the manuscript mentions scoRNN and Restricted-capacity uRNN have similar performance for 16k parameters and then state that scoRNN has the best test accuracy at 96.2%. However, there is no example for restricted-capacity uRNN with 69k parameters to show that the performance of restricted-capacity uRNN doesn't also increase similarly with more parameters.
-- Overall it’s unclear to me how to completely determine the benefit of this technique over the others because, for each of the tests, different techniques may have superior performance. For instance, LSTM performs best in 5.2 and in 5.3 for the MNIST test accuracy. scoRNN and Restricted-capacity uRNN perform similarly for permuted MNIST Test Accuracy in 5.3. Finally, scoRNN seems to far outperform the other techniques in table 2 on the TIMIT speech dataset. I don’t understand the significance of each test and why the relative performance of the techniques vary from one to the other.
-- For example, the manuscript seems to be making the case that the scoRNN gradients are more stable than those of a uRNN, but all of the results are presented in terms of network accuracy and not gradient stability. You can sort of see that generally the convergence is more gradual for the scoRNN than the uRNN from the training graphs but it'd be nice if there was an actual comparison of the stability of the gradients during training (as in Figure 4 of the Arjovsky 2016 paper being compared to for instance) just to make it really clear.

---

> ### Author Response · Authors · 2017-12-23
> **Review of "Orthogonal Recurrent Neural Networks with Scaled Cayley Transform"**
>
> Thank you for the comments.  Please see below.
> 1.)  “It’s not clear to me how D is determined for each test.”
> - It is not known a priori the optimal number of negative ones that should be included in the scaling matrix. In this work, the percentage of negative ones in the diagonal matrix D is considered a hyperparameter and is tuned for each experiment.  See Response to Reviewer 1 for details.
>
> 2.)  “For the copying and adding problem test cases,...make it clear why these experiments are relevant.”
> -This is a good point.  We have clarified why these experiments are useful.
>
> 3.)  “The ylabel in Figure 1 is “Test Loss” which I didn’t see defined.”
>  -The loss function is indeed cross-entropy.  This was clarified in the updated submittal.
>
> 4.)  “The plots in figure 1 and 2 have different colors to represent the same set of techniques.”
> - The figures have been modified to reflect this in the updated submittal.
>
> 5.)   “It looks like in Figure 1 the scoRNN is outperformed by the uRNN in the long run in spite of the scoRNN convergence being smoother, which should be clarified.”
>  -This has been noted in the updated submittal.
>
> 6.) “It looks like in Figure 2 the scoRNN is outperformed by the LSTM across the board, which should be clarified.”
> -This has been emphasized more in the updated submittal.
>
> 7.)   “How is test set accuracy defined in section 5.3?”
> - For the MNIST experiment, the test set accuracy is the percentage of digits in the test set that were classified accurately.  It has been clarified what is being tested in each experiment.
>
> 8.)   “When discussing table 1, the manuscript mentions scoRNN and Restricted-capacity uRNN have similar performance for 16k parameters and then state that scoRNN has the best test accuracy at 96.2%. However, there is no example for restricted-capacity uRNN with 69k parameters to show that the performance of restricted-capacity uRNN doesn't also increase similarly with more parameters.”
>  -In order to match the same number of parameters of approx. 69k, the restricted-capacity uRNN will require a hidden size of 2,170 units.  This is much larger than the hidden size of all the other models and we are not sure if we will be able to tune the machine in time but we are running a few experiments with this hidden size.
>
>
> 9.)  “Overall it’s unclear to me how to completely determine the benefit of this technique over the others...”
> -We agree that the results of each model on each task are not necessarily well-defined, and will add some description in Sections 5.1 and 5.2 to address this. Our intent was to show each model's performance in a variety of contexts where vanishing and exploding gradients occur. Although the scoRNN model does not always achieve the best results for each experiment, it is among the stronger performers and offers smoother convergence with smaller hidden states. Thus, the model is a competitive alternative to the LSTM model and the tested uRNNs.
>
> 10.)  “...stability of the gradients...”
>  -We have carried out experiments to compare gradient and hidden state stability of the LSTM and scoRNN models, similar to the figure in the 2016 Arjovsky paper. Our results show the norm of scoRNN hidden state gradients staying near constant at 10^-4 over 500 timesteps, while the LSTM gradient norm decays to 0 after 200-300 timesteps. These results have been included in a figure in the revised submittal in Appendix C.  We are having some difficulty in obtaining hidden state gradients from the full-capacity and restricted-capacity uRNNs, and will include these results if we are able.

---

> > ### Author Response · Authors · 2018-01-05
> > **Review of "Orthogonal Recurrent Neural Networks with Scaled Cayley Transform"**
> >
> > 8.)   “When discussing table 1, the manuscript mentions scoRNN and Restricted-capacity uRNN have similar performance for 16k parameters and then state that scoRNN has the best test accuracy at 96.2%. However, there is no example for restricted-capacity uRNN with 69k parameters to show that the performance of restricted-capacity uRNN doesn't also increase similarly with more parameters.”
> >
> > We have completed MNIST experiments for the restricted-capacity uRNN with 69k parameters and have included the results in Section 5.3 in the most recent paper revision. This machine's performance is comparable to the n=360 scoRNN on unpermuted MNIST and slightly worse than the n=360 scoRNN on permuted MNIST. The runtime for a single epoch was 50 minutes, which was nearly 7 times slower than the n=360 scoRNN, which also had approximately 69k parameters. We have updated Appendix D to include this information.

---

### Official Review · AnonReviewer1 · 2017-11-27
**An alternative parametrization of Unitary RNNs**

**Rating:** 6
**Confidence:** 3

**Review:**

This paper suggests an RNN reparametrization of the recurrent weights with a skew-symmetric matrix using Cayley transform to keep the recurrent weight matrix orthogonal. They suggest that they reparametrization leads to superior performance compare to other forms of Unitary Recurrent Networks.

I think the paper is well-written.  Authors have discussed previous works adequately and provided enough insight and motivation about the proposed method.

I have two questions from authors:

1- What are the hyperparameters that you optimized in experiments?

2- How sensitive is the results to the number of -1 in the diagonal matrix?

3- ince the paper is not about compression, it might be unfair to limit the number of hidden units in LSTMs just to match the number of parameters to RNNs. In MNIST experiment, for example, better numbers are reported for larger LSTMs. I think matching the number of hidden units could be helpful. Also, one might want to know if the scoRNN is still superior in the regime where the number of hidden units is about 1000. I appreciate if authors can provide more results in these settings.

---

> ### Author Response · Authors · 2017-12-23
> **An alternative parametrization of Unitary RNNs**
>
> Thank you for the comments.  Please see below.
>
> 1.)	“What are the hyperparameters that you optimized in experiments?”
> - For all methods, we tuned the optimizer type and learning rate. For scoRNN, we also tuned the percentage of  negative 1's on the diagonal D. See No. 2 below.
>
> 2.)  “How sensitive is the results to the number of -1 in the diagonal matrix?”
> - We tuned the percentage of -1s on the diagonal matrix first by multiples of 25% (i.e. 0, 25%, 50%, 75%, 100%) and then by multiples of 5% or 10%.  Tuning by multiplies of 5% and 10% did not usually affect the results significantly with very small differences when doing so.
>
> 3.) “Since the paper is not about compression, it might be unfair to limit the number of hidden units in LSTMs just to match the number of parameters to RNNs.”
> -We have increased the number of hidden units for the scoRNN and LSTM models on the MNIST experiment to 512, and will include the new results in Section 5.3 in the revised submittal. The results indicate little to no improvements for both of these models from the paper results and we suspect that increasing the hidden sizes further to 1000 will have no significant improvement.  We have also attempted to run LSTM with n=1000 but this turns out to be so slow in our system and uses up the entire GPU memory that we would not be able to tune the hyperparameters.  If we get results in time, we will include them in the paper.

---

> > ### Author Response · Authors · 2018-01-05
> > **An alternative parametrization of Unitary RNNs (cont)**
> >
> > 3.) “Since the paper is not about compression, it might be unfair to limit the number of hidden units in LSTMs just to match the number of parameters to RNNs.”
> >
> > As noted in the previous comment, we have been running the n=1000 LSTM but it will not finish completely by the deadline. However, partial results indicate that the n=1000 LSTM will not improve over the n=256 or n=512 LSTM results.

---

### Official Review · AnonReviewer3 · 2017-11-28
**Novel parametrization of RNNs allows representing orthogonal weight matrices relatively easily**

**Rating:** 5
**Confidence:** 4

**Review:**

The paper is clearly written, with a good coverage of previous relevant literature.
The contribution itself is slightly incremental, as several different parameterization of orthogonal or almost-orthogonal weight matrices for RNN have been introduced.
Therefore, the paper must show that this new method performs better in some way compared with previous methods. They show that the proposed method is competitive on several datasets and a clear winner on one task: MSE on TIMIT.

Pros:
1. New, relatively simple method for learning orthogonal weight matrices for RNN

2. Clearly written

3. Quite good results on several relevant tasks.

Cons:
1. Technical novelty is somewhat limited

2. Experiments do not evaluate run time, memory use, computational complexity, or stability. Therefore it is more difficult to make comparisons: perhaps restricted-capacity uRNN is 10 times faster than the proposed method?

---

> ### Author Response · Authors · 2017-12-23
> **Novel parametrization of RNNs allows representing orthogonal weight matrices relatively easily**
>
> Thank you for the comments.  Please see below.
>
> 1.)  “Technical novelty is somewhat limited.”
> - We believe that although there are several orthogonal RNNs, the scoRNN architecture has a new and much simpler scheme and is numerically more stable by maintaining orthogonality in the recurrent matrix.
>
> 2.)  “Experiments do not evaluate run time, memory use, computational complexity, or stability.”
> -We have carried out additional experiments to examine run time and the following results will be included in the revision.  The Full-Capacity uRNN and Restricted-Capacity uRNNs codes tested are from the Wisdom et al. 2016 paper and the LSTM is based on the builtin model in Tensorflow.  The results indicate that scoRNN is faster than the Full-Capacity uRNN and Restricted-Capacity uRNNs but slightly slower than Tensorflow's builtin implementation of LSTM.  We believe that the LSTM is faster because it is a builtin model that has been optimized within Tensorflow; there is virtually no change in runtime from hidden sizes 128 to 512.  Please see the table below for the unpermuted MNIST experiment.
>
>
> Model:          Hidden Size:          Approx. # Parameters         Time per Epoch(minutes):
> scoRNN               170                                 16k                                    5.3
> restr.-uRNN         512                                 16k                                    8.2
> full-uRNN            116                                16k                                   10.8
> LSTM                  128                                 68k                                    5.0
> scoRNN               360                                 69k                                    7.4
> scoRNN               512                              137k                                   11.2
> full-uRNN            360                             137k                                    25.8
> LSTM                  256                              270k                                     5.2
> full-uRNN           512                              270k                                   27.9
> LSTM                  512                           1,058k                                    5.6
>
> For memory usage and computational complexity, the scoRNN architecture is identical to a standard RNN except with the added storage for the n(n-1)/2 entries of the skew-symmetric matrix, and increased computational complexity from forming the recurrent weight matrix which is calculated once per training iteration.  This computational cost is small compared to the cost of the forward and backward propagations through all time steps and for all examples in a training batch. Thus, scoRNN’s complexity is almost identical to a standard RNN. The basic RNNs should compare favorably with other methods in complexity although we couldn’t find a reference for this.
>
>  As for stability, we assume the referee refers to the stability of the gradients with respect to hidden states during training (as in Figure 4 of the Arjovsky 2016) that referee 2 points out.  We have carried out experiments to compare gradient stability of the LSTM and scoRNN models with results indicating the scoRNN model has significantly more stable gradients than LSTM.  These results will be included in the revised submittal.  We are having some difficulties in obtaining hidden state gradients from the full-capacity and restricted-capacity uRNNs codes.
>
> These additions will be in two new appendices; stability will be addressed in Appendix C, and complexity & speed will be addressed in Appendix D.

---

### Decision · Program_Chairs · 2018-01-29
**ICLR 2018 Conference Acceptance Decision**

**Decision:**

Reject

**Comment:**

The authors use the Cayley transform representation of an orthogonal matrix to provide a parameterization of an RNN with orthogonal weights.   The paper is clearly written and the formulation is simple and elegant.  However,  I share the concerns of reviewer 3 about the significance of another method for parameterizing orthogonal RNN, as there has not been a lot of evidence that these have been useful on real problems (and indeed, on most of the toys used show the value of orthogonal RNN, one can get good results just by orthogonal initialization, e.g. as in Henaff et. al. as cited in this work).    This work does not compare experimentally against many of the other methods, e.g. https://arxiv.org/pdf/1612.00188.pdf,  the two Jing et. al. works cited, simple projection methods (either full projections at each step or stochastic projections as in Henaff et. al.).  It does not cite or compare against the approach in https://arxiv.org/pdf/1607.04903.pdf.